# Novel Machine Learning Approach to Predict and Personalize Length of Stay for Patients Admitted with Syncope from the Emergency Department

**DOI:** 10.3390/jpm13010007

**Published:** 2022-12-20

**Authors:** Sangil Lee, Avinash Reddy Mudireddy, Deepak Kumar Pasupula, Mehul Adhaduk, E. John Barsotti, Milan Sonka, Giselle M. Statz, Tyler Bullis, Samuel L. Johnston, Aron Z. Evans, Brian Olshansky, Milena A. Gebska

**Affiliations:** 1Department of Emergency Medicine, Carver College of Medicine, University of Iowa, 200 Hawkins Drive, Iowa City, IA 52242, USA; 2The Iowa Initiative of Artificial Intelligence, University of Iowa, 103 South Capitol Street, Iowa City, IA 52242, USA; avinashreddy-mudireddy@uiowa.edu; 3Division of Cardiology, Mercy One North Iowa Heart Center, 250 S Crescent Dr, Mason City, IA 50401, USA; 84pdeepak@gmail.com; 4Department of Internal Medicine, Carver College of Medicine, University of Iowa, 200 Hawkins Drive, Iowa City, IA 52242, USA; mehul-adhaduk@uiowa.edu (M.A.); tyler-bullis@uiowa.edu (T.B.); aron-evans@uiowa.edu (A.Z.E.); 5Department of Epidemiology, College of Public Health, University of Iowa, 145 N. Riverside Drive, Iowa City, IA 52242, USA; john-barsotti@uiowa.edu; 6Division of Cardiovascular Medicine, Carver College of Medicine, University of Iowa, 200 Hawkins Drive, Iowa City, IA 52242, USA; giselle-statz@uiowa.edu (G.M.S.); samuel-johnston@uiowa.edu (S.L.J.)

**Keywords:** syncope, length of stay, artificial intelligence, machine learning, prediction

## Abstract

**Background**: Syncope, a common problem encountered in the emergency department (ED), has a multitude of causes ranging from benign to life-threatening. Hospitalization may be required, but the management can vary substantially depending on specific clinical characteristics. Models predicting admission and hospitalization length of stay (LoS) are lacking. The purpose of this study was to design an effective, exploratory model using machine learning (ML) technology to predict LoS for patients presenting with syncope. **Methods**: This was a retrospective analysis using over 4 million patients from the National Emergency Department Sample (NEDS) database presenting to the ED with syncope between 2016–2019. A multilayer perceptron neural network with one hidden layer was trained and validated on this data set. **Results**: Receiver Operator Characteristics (ROC) were determined for each of the five ANN models with varying cutoffs for LoS. A fair area under the curve (AUC of 0.78) to good (AUC of 0.88) prediction performance was achieved based on sequential analysis at different cutoff points, starting from the same day discharge and ending at the longest analyzed cutoff LoS ≤7 days versus >7 days, accordingly. The ML algorithm showed significant sensitivity and specificity in predicting short (≤48 h) versus long (>48 h) LoS, with an AUC of 0.81. **Conclusions**: Using variables available to triaging ED clinicians, ML shows promise in predicting hospital LoS with fair to good performance for patients presenting with syncope.

## 1. Introduction

Syncope, one of the most common conditions seen in medical practice, is a major cause for emergency department (ED) visits. Between 13–83% of such patients are hospitalized [1]. Variation in admission rates reflects the complexity of the problem and lack of evidence-based consensus on effective syncope assessment criteria for hospital admission [1,2]. Some etiologies of syncope are easily identified and benign, whereas others are difficult to determine. In particular, cardiogenic causes and other high-risk conditions can be difficult to identify in the ED. Often extensive, expensive, unnecessary, and potentially harmful evaluations are undertaken due to fear of missing life-threatening or otherwise serious underlying causes. Such approaches consume hospital resources and often lead to prolonged and unnecessary hospital length of stay (LoS) [1,3,4,5,6]. Long initial hospital stays (>3 days) are associated with greater expense, higher readmission rates within 30 days and resource utilization [7].

LoS is a metric that can determine hospitals’ triage and illness severity, assess overall healthcare cost, and identify resource allocation regarding staff and patient needs [8]. Hospitals in the United States and worldwide rely on Diagnosis Related Groups (DRG) and Utilization Management services to reduce unnecessary and prolonged hospitalizations. Short-term observation units help decrease LoS; however, the overall healthcare cost related to syncope continues to rise [9]. Patient-specific, personalized, computer-generated alerts to admitting physicians giving the mean target LoS based on a provisional DRG assignment Interestingly, despite decades of experience, advances in diagnostic techniques, and implementation of variouswas associated with approximately 3% reduction in mean LoS [10]. Interestingly, despite decades of experience, advances in diagnostic techniques, and implementation of various syncope risk stratification algorithms, a reliable benchmark LoS prediction for patients presenting to the ED with syncope remains elusive.

This study aimed to design an effective and exploratory model to help ED physicians predict LoS for each patient presenting with syncope. We hypothesized that machine learning (ML) technology can be trained effectively to make such personalized prediction using retrospective data. Creating predictive instruments using an artificial neuronal network (ANN) has been previously validated in different clinical settings [8,11], and has been shown to predict LoS in intensive care units after cardiac surgery [12] or craniotomy [13]. Using a similar approach, we subjected a set of input variables to a series of computer training to predict which syncope patients would qualify for an observation unit (predicted hospital stay <48 h) versus an extended inpatient stay (>48 h). Such a personalized and patient-centered approach could significantly impact quality of care and promote better hospital resource utilization. Further, creating an effective point-of-care risk stratification tool could help ED physicians and other healthcare providers identify patients at high risk for prolonged hospitalization, death, and other adverse outcomes. 

## 2. Materials and Methods

### 2.1. Data Source

This retrospective analysis is from the National Emergency Department Sample (NEDS), the largest national all-payer ED dataset in the US. The dataset is developed by the Healthcare Cost and Utilization Project (HCUP) and is publicly available. After applying appropriate discharge weights, the estimates from this data set yield national estimates that have been previously validated [14,15,16]. NEDS is compiled annually from the nationwide EDs and inpatient databases, capturing 68.7% of the total population and 78.2% of all ED visits from 37 geographically dispersed states in the US. 

NEDS provides de-identified, patient-specific, and limited information about the encounter in the ED and subsequent inpatient care, if admitted. These details are compiled at the end of hospitalization in the form of the International Classification of Disease, Tenth Revision, Clinical Modification (ICD-10-CM) diagnosis and procedure codes. The dataset was analyzed in compliance with the Health Insurance Portability and Accountability Act (HIPAA) of 1996. Therefore, the study was exempt from institutional review board approval. 2.2. Study Overview and Participants

We included all patients who presented to the ED primarily with syncope. Syncope was defined as all encounters with a primary discharge ICD-10-CM code diagnosis of R55 (Syncope and Collapse). Utilization of the ICD-10 code to diagnose syncope in an administrative database has been externally validated with 63% sensitivity and has a 99.5% positive predictive value [17]. To maintain uniformity in ICD codes, we limited our study to years with the ICD-10 codes in the NEDS dataset. Therefore, we analyzed the dataset from 1 January 2016 to 31 December 2019. We excluded patients under age 18 and those without mortality data.

Database contains patient demographics such as patient age, gender, and race (calendar year 2019 only), primary insurance provider, and hospital demographics including hospital ownership, teaching status, urban-rural designation, and trauma level is provided in the dataset. Hospitalization-specific patient characteristics including diagnoses (acute and chronic), procedures performed, disposition from the ED, and inpatient admission information is also available.

### 2.2. Input Variables

To maximize the predictive capability of our model, we included patient factors such as age, sex, race, and ethnicity, as well as 31 Elixhauser comorbidity indices (ECI) computed from ICD-10-CM diagnosis codes which were utilized in the ML algorithm as a representation of personalized cardiovascular risk factors and include: AIDS/HIV, alcohol abuse, blood loss anemia, cardiac arrhythmias, chronic pulmonary disease, coagulopathy, congestive heart failure, deficiency anemia, depression, diabetes-complicated, diabetes-uncomplicated, drug abuse, fluid-electrolyte disorder, hypotension, hypertension-complicated, hypertension-uncomplicated, hypothyroidism, liver disease, lymphoma, metastatic cancer, obesity, other neurological disorder, paralysis, peptic ulcer disease excluding bleeding, peripheral vascular disorders, psychoses, pulmonary circulation disorder, renal failure, rheumatoid arthritis-collagen vascular disorder, stroke, solid tumor without metastasis, valvular disease and weight loss. 

ECI is a validated method for categorizing patient specific comorbidities in a large administrative database based on ICD diagnosis codes [18,19]. The larger the ECI score, the higher the comorbidity burden. We chose the ECI over individual ICD-10-CM codes to provide a personalized approach and reduce complex dimensionality that would have been created from a unique ICD-10 code.

All the 31 ECIs were computed if the pertinent ICD-10-CM diagnosis code was present for an individual patient; this included all the subcategories for the same disease. If none of the ICD-10 codes were present, then that comorbid condition was considered absent. Once all the 31-ECI were computed, a composite sum ECI score was calculated after applying predefined individual comorbidity weights [18]. The sum ECI score is a continuous variable that has a positive skewed (right skewed) distribution; with nearly 36% of patients having a sum ECI score of 0 (i.e., lacking any of the 31-comorbid condition). In order to provide a clinically meaningful information, we clustered the sum ECI score (continuous variable) into a categorical variable with 3 levels: 0 (no sum ECI), 1 to 2 (one or two sum ECI score), and > 3 (three or more sum ECI score). ECI was computed utilizing a predefined code in the statistical package used in this study. ECI variables of the study population have been summarized in the Appendix A.

Age (continuous variable) was also clustered into 4 subgroups: 18–40, 40-60, 60-75, and ≥75 years. As for facility-level factors, we included rurality based on metropolitan statistical area (MSA) status and whether the ED is affiliated with an academic institution or community hospital. In addition, the ED encounter during weekends versus weekdays was also analyzed. After converting the aforementioned input variables to indicator variables, a total of 72 variables were included. 

### 2.3. Target Variables (Study Outcomes)

The primary outcomes of interest included LoS among syncope patients and death in ED or during inpatient hospitalization. We categorized LoS into short stay (negative class) and extended stay (positive class). Five models considering different short/long separation cutoffs of LoS were created. Short versus extended stay models were ≤0 days (indicating ED discharge without hospitalization) versus >0 days, ≤24 h versus >24 h, ≤48 h versus >48 h, ≤4 days versus >4 days, and ≤7 days versus > 7 days. In-hospital mortality was considered a competing outcome for hospital LoS and it was categorized as extended stay.

### 2.4. Statistical Analysis

First, we reported the descriptive statistics for the syncope cohort between 2016–2019. Then, we introduced the study sample into an ANN algorithm to develop and validate the outcome of death or extended LoS using the input variables described above. Metrics reported include prediction performance precision, recall, F1 score, and accuracy for each of the models built. Receiver-operating characteristics (ROC) and the area under the ROC curve (AUC) were evaluated and plotted. 

Consider a task of predicting “short” LoS. The metrics used to evaluate the predictive performance of trained classifiers are defined as follows. True positive (TP) decisions reflect correct predictions (e.g., a patient whose LoS was predicted as “short” and was indeed discharged earlier than the LoS cutoff). True negative (TN) decisions also represent correct predictions (e.g., a patient whose LoS was predicted as “long” and was indeed discharged later than the LoS cutoff). False positive (FP) and false negative (FN) predictions are defined accordingly (e.g., a false positive decision reflects a “short” LoS prediction for a patient who was discharged later than the LoS cutoff). 

Precision (or positive predictive value) is the ratio of the number of correct TP decisions and the sum of TP + TN decisions (Precision = TP/(TP + FP). Recall (or sensitivity) is the ratio of the number of correct TP decisions and the sum of TP + FN decisions (Recall = TP/(TP + FN). F1 score is a mean of Precision and Recall, calculated as F1 = 2TP/(2TP + FP + FN), to reflect both the precision and recall in a single combined performance metric. Accuracy is a ratio of the number of all correct decisions vs. the number of all decisions (Accuracy = (TP + TN)/(TP + TN + FP + FN). Finally, a receiver operating characteristic (ROC) curve is a graphical plot that illustrates the diagnostic ability of a binary classifier system as its discrimination threshold is varied. The ROC curve is created by plotting the Recall against the rate of FP decisions (probability of false decision or 1-specificity) at various operating threshold settings. The closer the area under the ROC curve is to one (AUC = 1), the better classifier performance is observed for the assessed prediction (decision-making) system.

Patient demographics, ECI, and hospital demographics were stratified by the calendar year and compared using the Pearson chi-squared test after accounting for the complex survey design of NEDS. As per the HCUP data reporting policy, any variable with *n* < 10 is not presented. A two-sided *p* < 0.05 was considered statistically significant. All analyses were performed using IM SPSS Statistics, version 27.0 (IBM Corp, IL, USA), STATA version 16.1 (Stat Corp. LP, TX, USA), and Python 3.8. 

### 2.5. Data Processing and Machine Learning

ANNs are modeled by mimicking neurons in the brain. In biological systems, the learning happens through adjustments to synaptic connections between the neurons. Similarly, in an ANN model, this happens through updating the weights of the connections between the nodes/neurons. Multilayer perceptron (MLP) networks are feedforward ANNs that contain at least three layers: an input, a hidden, and an output layer. The learning happens through an iterative, supervised mechanism where the information passes through input, hidden, and output layers to predict an outcome. During the subsequent learning epochs, the predicted outcome is compared with the ground truth to determine the ANN’s performance—the prediction error at each epoch. For each next learning approach, the weights between the layers are updated in a back propagation manner to reduce the error in the subsequent iteration. This process is repeated until the error is minimized through this backpropagation optimization process.

As described in Section 2.3, we used a 72-dimensional input to predict a binary outcome: short stay or extended stay. Several cutoff thresholds of what constituted an extended (long) stay were identified and predictive ANN performance was tested. We used upsampling and stratified sampling techniques to address the class imbalance in the dataset. The dataset is further divided into a train-validation-test split of 64:16:20. We evaluated the parameter tuning during the development of the ANNs. 

Our neural network (Figure 1) has three hidden layers with 64, 32, and 16 neurons fired by rectified linear unit activation. Each layer is further treated with batch normalization and dropouts (0.05). The output layer has a binary outcome with “sigmoid” activation. We used Adam optimizer (lr = 0.001), the binary cross entropy loss as optimizer, and loss function, respectively. The model is trained for 750 epochs with 8192 as batch size. We divided 64% of the entire samples into a training, 16% for validation, and 20% for test. 

## 3. Results

### 3.1. Participant Characteristics

We identified a total of 4,645,483 patient presentations to EDs for syncope in the United States between 2016–2019 and this entire set of almost 5 million patients was used in this study. The 18 to 54 years of age group consistently represented nearly half of the population. Likewise, over half the study population was women and over half were insured by Medicare or Medicaid. Demographic details of the study population after stratifying based on the calendar year are presented in Table 1. A total of 929 (<0.01%) died in the ED or hospital. Appendix A provide additional information related to patient and hospital characteristics.

### 3.2. Prediction Model Performance for Short and Long LoS after Syncope

As stated in Section 2.4, we built five models with various target variable cutoffs for hospital LoS. The results reflecting the prediction performance for all the models are given in Table 2.

### 3.3. The Receiver Operator Characteristics for Short versus Long LoS

Receiver Operator Characteristics (ROC) were determined for each of the five ANN models. As depicted in Figure 2, a fair (AUC of 0.78) to good (AUC of 0.88) prediction performance was achieved based on sequential analysis at different cutoff points, starting from the shortest LoS ≤0 days (indicating ED discharge or same day discharge from the hospital) versus >0 days (Figure 2a) and ending at the longest analyzed cutoff LoS ≤7 days versus >7 days (Figure 2e), accordingly.

## 4. Discussion

Syncope is a symptom with multiple causes and explicit complexity in management. Hospitalization to facilitate diagnostic and therapeutic interventions depends on patient presentation and comorbidities. An accurate prediction of LoS in patients with syncope would be useful in terms of facilitating patient care and allocating hospital resources. However, it is difficult to determine a patient’s need for hospitalization, let alone their ideal LoS even after the rigorous application of available guidelines [1,4,5], physician intuition, a careful history, a physical examination including orthostatic blood pressures, obtaining a 12-lead ECG and other routine tests available in the ED.

Prior studies have demonstrated significant variations in LoS among patients hospitalized for syncope depending on individual physician preference, hospital, and geographic region [20,21]. To date, no simple clinical calculator has proven effective in predicting LoS in syncope patients. ML is a potentially attrFactive tool to predict LoS in these patients, as well as other outcomes, because it can identify non-intuitive patterns from extremely large datasets, such as the NEDS. 

ML has previously been used in patients with inflammatory bowel disease to accurately identify high-need, high-cost patients and LoS [20], as well as ICU mortality and LoS [21]. Similarly, virtual elements of artificial intelligence can predict LoS in patients undergoing total knee arthroplasty [22], lumbar decompression surgery [23], craniotomy clipping, [24] and other conditions. 

ANN, a type of ML that mimics the human brain, is capable of handling very complex tasks derived from non-linear elements. Here, we show that ANN shows potential to predict LoS for patients with syncope, and for the great majority of these patients, the LoS will be “short”. Using the NEDS dataset, we can predict short or ≤48 h LoS in syncope patients with an AUC of 0.81, compared to long (>48 h) LoS. Ultimately, for such a test to be employed clinically it would need to perform with an AUC of 0.90 or higher. 

Various explanations exist to potentially explain deficiencies in our prediction model. For example, it is likely that certain variables that are critical to predicting LoS do not exist within the NEDS dataset, or other national retrospective data sets. Including additional data, such as results from clinical testing and results from longitudinal follow-up would likely make this ML algorithm even more predictive and may also have the potential to reveal other clinical insights. Prospective studies will be necessary to test these hypotheses. 

Further roles for artificial intelligence in the evaluation of patients with syncope are to be expected. Determining outcomes based on initial clinical characteristics will likely translate into better understanding of this complex symptom, better patient outcomes, and better outcome measures. Understanding the relationships between syncope, underlying medical conditions, and risk factors will also remain a focus. These issues continue to be major challenges in the assessment of patients with syncope, and despite great efforts, progress has been slow to curb unnecessary hospitalizations and provide better methodologies for diagnostic assessments and utilization of hospital admissions.

## 5. Study Limitations

This study is limited by information available in the NEDS dataset. The analysis is retrospective and the LoS is not validated as being what is required for patient care. Nevertheless, these data represent how most US physicians act under the circumstance. The dataset assumes accurate coding and diagnoses, and lacks important clinically relevant data (e.g., results from clinical testing). It is also limited to a single patient encounter, so information regarding re-admission rates is unknown and longitudinal follow-up is unavailable. As the dataset is created at the end of the hospital encounter, we were not able to differentiate whether the comorbidity diagnosis was made at the initial encounter or during the hospitalization. 

The lack of long term follow-up makes identifying relevant, low-frequency events, like mortality, unhelpful as a classifying feature or outcome. Finally, a patient’s LoS is affected by medical and nonmedical factors that cannot be fully discerned from the present data. Whereas other studies have analyzed LoS for a relatively finite number of diagnoses [12], syncope is a heterogenous diagnosis with numerous and varied etiologies. The variation in LoS in this population may lead to confounding bias, again, not necessarily expressed in ICD-10 codes. 

## 6. Conclusions

Syncope, due to the diversity of causes, clinical presentations, and underlying comorbidities, remains a challenging and potentially costly problem to manage. Using a large national database with routinely available clinical parameters, we applied a novel ML approach to show promise in predicting hospital LoS for patients presenting with syncope with fair to good performance in AUC ranging from the same day discharge, short, to long LOS. Challenges remain in improving the predictive accuracy and assessing the need for hospitalization. Further analyses can help identify which variables are critical in the model and determine how these variables can predict a multiplicity of outcomes aside from LoS in a patient-centered, practical, efficient, and clinically relevant manner. 

With rapid advancements in personalized medicine and deep neural networks, exciting opportunities exist in the prediction models that would help clinicians understand and manage complex clinical decisions, such as syncope.

## Figures and Tables

**Figure 1 jpm-13-00007-f001:**
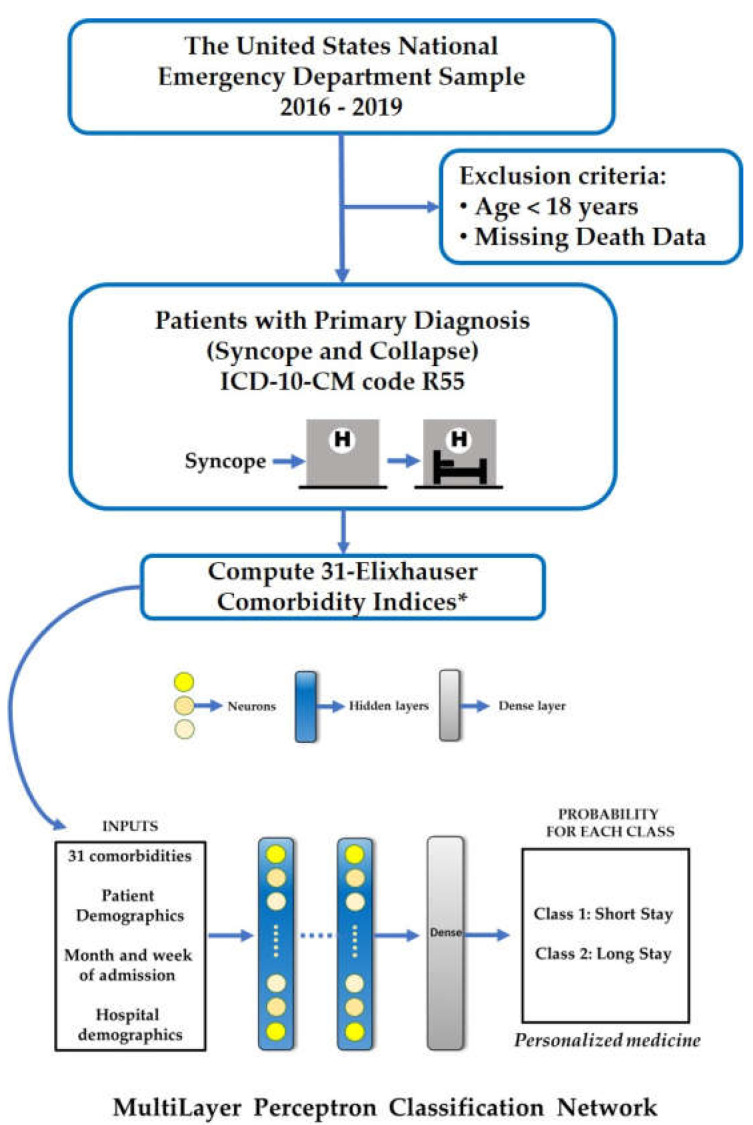
Study Design. * Uncomplicated hypertension, Cardiac arrhythmias, Fluid and electrolyte disorders, Uncomplicated diabetes, Chronic pulmonary disease, Complicated hypertension, Hypothyroidism, Renal failure, Depression, Congestive heart failure, Complicated diabetes, Neurological disorders, Obesity, Valvular disease, Peripheral vascular disorders, Drug abuse, Alcohol abuse, Rheumatoid arthritis/collagen vascular disorder, Solid tumor without metastasis, Deficiency anemia, Coagulopathy, Pulmonary circulation disorder, Liver disease, Psychoses, Weight loss, Metastatic cancer, Lymphoma, Paralysis, Peptic ulcer disease excluding bleeding, AIDS/HIV and Blood loss anemia.

**Figure 2 jpm-13-00007-f002:**
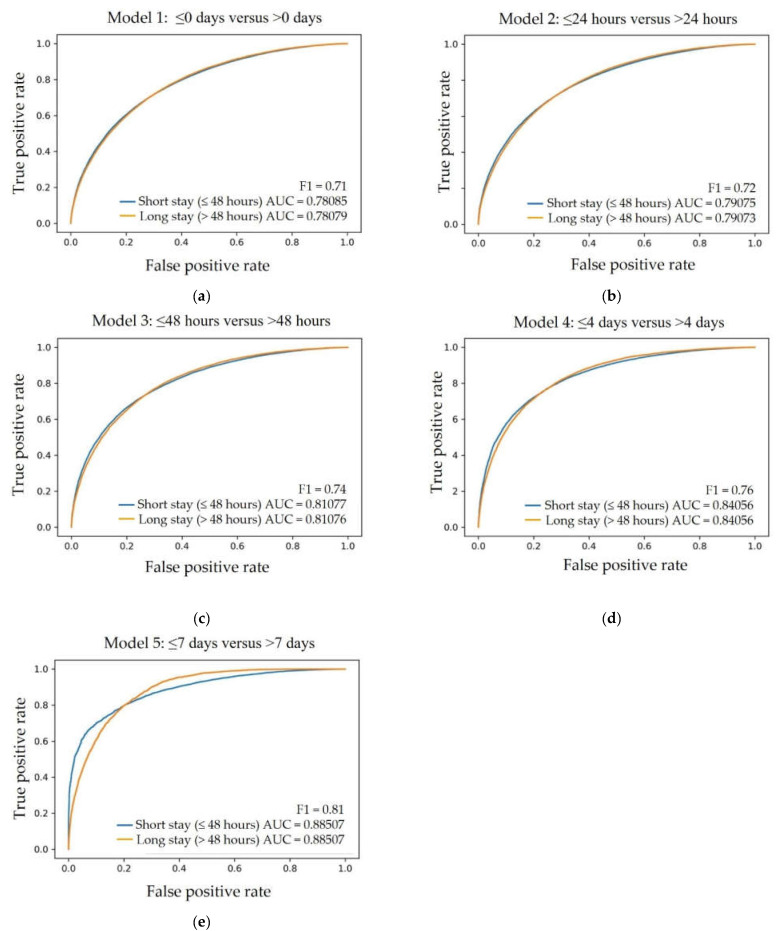
Receiver Operator Characteristics (ROC), corresponding AUC, and F1 score values are given for each of the five LoS prediction models: (**a**) ≤0 days (indicating ED discharge) versus >0 days, (**b**) ≤24 h versus >24 h, (**c**) ≤48 h versus >48 h, (**d**) ≤4 days versus >4 days, (**e**) ≤7 days versus >7 days.

**Table 1 jpm-13-00007-t001:** Patient characteristics.

	Total (*n* = 4,645,483)		2016 (*n* = 1,135,359)		2017 (*n* = 1,174,452)		2018 (*n* = 1,137,276)		2019 (*n* = 1,188,396)		P _trend_
Age (clustered)											
18–54 years	2,108,750	(45.4%)	533,875	(46.6%)	528,642	(45.0%)	516,383	(45.4%)	529,850	(44.6%)	<0.001
55–64 years	693,974	(14.9%)	167,632	(14.6%)	175,699	(15.0%)	171,566	(15.1%)	179,077	(15.1%)	
65–74 years	727,565	(15.7%)	171,271	(15.0%)	184,783	(15.7%)	178,412	(15.7%)	193,099	(16.2%)	
75–84 years	668,526	(14.4%)	160,514	(14.0%)	169,815	(14.5%)	163,382	(14.4%)	174,814	(14.7%)	
≥85 years	446,668	(9.6%)	112,067	(9.8%)	115,513	(9.8%)	107,532	(9.5%)	111,556	(9.4%)	
Gender											
Males	2,042,422	(44.0%)	498,284	(43.5%)	514,575	(43.8%)	499,703	(43.9%)	529,860	(44.6%)	<0.001
Females	2,602,663	(56.0%)	646,853	(56.5%)	659,848	(56.2%)	637,502	(56.1%)	658,460	(55.4%)	
ECI Cluster											
ECI = 0	1,670,288	(36.0%)	426,332	(37.2%)	421,696	(35.9%)	403,301	(35.5%)	418,959	(35.3%)	0.0043
ECI = 1–2	1,905,336	(41.0%)	470,893	(41.1%)	482,367	(41.1%)	465,062	(40.9%)	487,014	(41.0%)	
ECI ≥ 3	1,069,859	(23.0%)	248,135	(21.7%)	270,389	(23.0%)	268,913	(23.6%)	282,423	(23.8%)	
Primary expected payer											
Medicare	1,877,544	(40.5%)	458,255	(40.0%)	479,524	(40.9%)	457,512	(40.3%)	482,252	(40.6%)	0.1328
Medicaid	679,442	(14.6%)	169,790	(14.8%)	170,883	(14.6%)	171,253	(15.1%)	167,515	(14.1%)	
Private insurance	1,514,372	(32.6%)	376,475	(32.9%)	377,400	(32.2%)	370,989	(32.7%)	389,507	(32.8%)	
Self-pay	393,866	(8.5%)	94,992	(8.3%)	99,525	(8.5%)	95,935	(8.4%)	103,413	(8.7%)	
No charge	14,042	(0.3%)	3299	(0.3%)	3,948	(0.3%)	2864	(0.3%)	3931	(0.3%)	
Other	159,652	(3.4%)	41,594	(3.6%)	40,110	(3.4%)	37,654	(3.3%)	40,293	(3.4%)	
Death/Alive											
Alive	4,643,245	(100.0%)	1,144,608	(99.9%)	1,173,888	(100.0%)	1,136,846	(100.0%)	1,187,904	(100.0%)	0.0027
Died in ED	1309	(<0.01%)	464	(<0.01%)	330	(<0.01%)	227	(<0.01%)	288	(<0.01%)	
Died in the Hospital	929	(<0.01%)	287	(<0.01%)	235	(<0.01%)	203	(<0.01%)	204	(<0.01%)	

ECI = Elixhauser Comorbidity Index cluster, computed from the weighted sum ECI score which in calculated from the thirty-one individual comorbidities.

**Table 2 jpm-13-00007-t002:** The predictive performance of hospital length of stay.

Length of Stay	AUC *	Precision	Recall	F1	Average Accuracy
≤0 days #	0.78	0.70	0.72	0.71	0.71
≤24 h	0.79	0.72	0.72	0.72	0.72
≤48 h	0.81	0.72	0.76	0.74	0.73
≤4 days	0.84	0.76	0.75	0.76	0.76
≤7 days	0.88	0.78	0.83	0.81	0.80

* AUC—Area Under the Curve. # <0 days indicates discharge from ED. The AUC for each cutoff point ranged from “fair” to “good”. In order to reflect the most pressing clinical relevance, the LoS cutoff value of ≤48 h versus > 48 h was considered to be the most practical in the real-world setting (Figure 2c—AUC 0.81; precision 0.72; recall 0.76; F1 0.74, average accuracy 0.73).

## Data Availability

NEDS is a publicly available deidentified database with restricted access to only those who have complete data usage training with the Health Care Utilization Project (HCUP). Therefore, we are unable to share the study data. Python code is available in https://github.com/Avi1122/ANN-code-for-NEDS-syncope-2022 (accessed on 28 September 2022).

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
