# Peer review of "Novel Machine Learning Approach to Predict and Personalize Length of Stay for Patients Admitted with Syncope from the Emergency Department"

_jpm, 2022, doi:10.3390/jpm13010007_

Round 1

Reviewer 1 Report

The manuscript presents an analysis of over four million patients with syncope from the National Emergency Department Sample (NEDS) database with presentations between 2016-2019. A multilayer perceptron network (MPN or ANN) was used to predict length of stay at several binary cutoffs based on > 0 days, > 24 hours, > 48 hours, > 4 days, and > days. The best AUC and F1 score were obtained for the latter two cutoffs: 0.84 and 0.76 for > 4 days. and 0.88 and 0.81 for > 7 days.  The manuscript is well-written. I have the following suggestions for improvement.

1. The multilayer perceptron network is essentially a collection of single perceptrons with feed-forward processing, and single perceptron is equivalent to logistic regression because multiple scalar parameters are used to predict a binary outcome with a single perceptron. What was the rationale for using an ANN/MPN in this problem instead of logistic regression? Please report the AUCs that would have been obtained with standard logistic regression to demonstrate whether the ANN approached used in the paper has an advantage over standard logistic regression. In the logistic regression model, please report the parameters for the full model, including adjusted odds ratios and p-values for all covariates.

2. The authors mention 31 Elixhauser comorbidity "indices," but I think they mean 31 "categories" in the ECI reported in Garland A, Fransoo R, Olafson K, Ramsey C, Yogendran M, Chateau D, McGowan K. The Epidemiology and Outcomes of Critical Illness in Manitoba. Winnipeg, MB: Manitoba Centre for Health Policy, 2012. Please state whether this was intended and make the correction. All 31 categories should be listed in abbreviated form in the main body of the manuscript, as the reader should not have to look to the references to figure out which 31 predictors were used. Is it possible to determine the relative contributions of these categories and other variables? Could a similar performance be obtained with a more parsimonious set of predictors?

3. Some of the continuous variables like age were converted to categorical variables. Continuous variables can be accommodated easily with logistic regression. Please also report whether a logistic regression model without converting continuous variables to categorical variables would perform as well as logistic regression models with use of categorical predictors only.

4. Please provide some additional comments regarding the accuracy of the administrative predictors available in this database.

5. Will the ANN be made available to readers so they can predict LOS for their own patients with syncope? This would be important to help make the results useful to others.

6. Usually there is a training/validation/testing split in these analyses, such as 70%/15%/15%. This should be reported.

Reviewer 2 Report

 A good attempt by the authors who choose a common problem which is a need of this hour.   Overall well-written manuscript with all the required information. Though, the following are my comments which can be incorporated before the final publication.
  • ECI score (con-129 continuous variable) into a categorical variable with 3 levels: 0 (no sum ECI), 1 to 2 (one or 130 two sum ECI score), and > 3 (three or more sum ECI score). This ECI Score can be represented in tabular form.
  • Name of the statistical package used in this study?
  • Results: The summary of the result in the abstract is written well from the reader's point of view, but it is missing in the main result section of the manuscript. Though figures and tables are there in the section results should be explained more in detail from the score point of view and their relevance could be explained in the discussion section in greater detail. This is the major thing that is lacking in the manuscript.
  • Even the conclusion in the abstract and main manuscript should have the same meaning and be in greater detail in the main manuscript.
  • The Result and conclusion should have some measurable information in the main section of the manuscript like the abstract which makes this article more informative from readers' point of view.
Thank you.
